# Mixed-Culture Metagenomics of the Microbes Making Sour Beer

Renan Eugênio Araujo Piraine [1,2], Fábio Pereira Leivas Leite [2] and Matthew L. Bochman [1,3,*]

1 Bochman Lab, Department of Molecular and Cellular Biochemistry, Indiana University, Bloomington, IN 47405, USA; renanbiotec@gmail.com
2 Laboratório de Microbiologia, Centro de Desenvolvimento Tecnológico, Universidade Federal de Pelotas, Pelotas 96010-610, RS, Brazil; fabio_leite@ufpel.edu.br
3 Wild Pitch Yeast, Bloomington, IN 47405, USA
* Correspondence: bochman@iu.edu; Tel.: +1-812-856-2095

**Abstract:** Mixed microbial cultures create sour beers but many brewers do not know which microbes comprise their cultures. The objective of this work was to use deep sequencing to identify microorganisms in sour beers brewed by spontaneous and non-spontaneous methods. Twenty samples were received from brewers, which were processed for microbiome analysis by next generation sequencing. For bacteria, primers were used to amplify the V3-V4 region of the 16S rRNA gene; fungal DNA detection was performed using primers to amplify the entire internal transcribed spacer region. The sequencing results were then used for taxonomy assignment, sample composition, and diversity analyses, as well as nucleotide BLAST searching. We identified 60 genera and 140 species of bacteria, of which the most prevalent were *Lactobacillus acetotolerans*, *Pediococcus damnosus*, and *Ralstonia picketti/mannitolilytica*. In fungal identification, 19 genera and 26 species were found, among which the most common yeasts were *Brettanomyces bruxellensis* and *Saccharomyces cerevisiae*. In some cases, genetic material from more than 60 microorganisms was found in a single sample. In conclusion, we were able to determine the microbiomes of various mixed cultures used to produce beer, providing useful information to better understand the sour beer fermentation process and brewing techniques.

**Keywords:** microbiome; mixed-fermentation; sour; beer; yeast; bacteria





## 1. Introduction

Traditionally, beer production methods are divided into two categories: (a) fermentation at the bottom of the fermenter, performed by *Saccharomyces pastorianus* in lager beer production, and (b) top-fermented beers, in which *S. cerevisiae* yeast ferments at the top of the wort, producing ales. Extending the concept to mixed fermentations, two new categories can be included: (c) non-spontaneous fermentation, carried out by an in-house starter culture which consists of yeast and lactic acid bacteria (LAB), and (d) spontaneous fermentation, in which microorganisms such as enterobacteria, yeasts, mold, LAB, and acetic acid-producing bacteria (AAB), among others, are inoculated through exposure to ambient air or external sources (e.g., wood, flowers, or fruits) to ferment these beers [1,2]. Moreover, wooden barrels and foeders used during fermentation act as additional source for microbial inoculation in beer wort [3–5].

Mixed fermentations are generally performed by yeasts and LAB in the process of creating sour beers, forming a complex microbiome that acts through their interaction and cooperation [6]. Microorganisms and their enzymes are used through biotechnological processes for acidification, alcohol production, proteolysis, lipolysis, and amino acid conversion in beer wort [1]. Various phases of fermentation can be identified in mixed fermentation beers, in which different bacteria and yeasts are isolated at specific periods [7], including novel microorganisms not yet characterized [5]. Changes in the presence

and concentrations of various microorganisms suggest the existence of a microenvironment regulated according to substrate conversion and growth-limiting factors such as pH, carbohydrate concentration, oxygen, temperature, and alcohol concentration [8].

The global beer market is experiencing a resurgence in the interest in sour beers because new products and more complex flavors are being obtained by large and small production breweries around the world [4,9]. Non-spontaneous and spontaneous mixed fermentations are processes used by brewers worldwide to produce these sour beers [4]. However, most brewers do not know exactly which microorganisms are present in their mixed cultures, as well as are unaware of the relative proportions of these microbes in their starter cultures. Even though wild yeasts, LAB, and some Gram-positive bacteria are considered contaminants in the vast majority of beer fermentations, these same microorganisms are often highly desired for the production of specific sour and wild beer styles. As an example, Lambics and American Coolship Ales are beverages with unique flavor profiles generated by "spoilage" organisms [10], in which dozens of volatile compounds can be identified as a direct result of the microbial interactions and release of fermentation by-products [11]. Thus, it is of interest to know the microorganisms present in these fermentative processes, seeking to characterize the microbial diversity and parameters that influence sensory perceptions, and to deepen the knowledge of mixed fermentation beers. The objective of this study was to identify bacteria and yeasts present in different beers and mixed-culture samples from several locations produced by homebrewers and craft breweries using spontaneous and non-spontaneous fermentation methods.

## 2. Materials and Methods

### 2.1. Samples

The samples used in this study were obtained through the crowdfunding project "Mixed culture metagenomics of the microbes making sour beer" (DOI 10.18258/13495) hosted on the Experiment platform (www.experiment.com, accessed on 2 July 2021). Each brewer collected and sent their own sample in glass beer bottles or plastic vials containing beer or slurry, which were stored at 4 °C until their analysis.

During July–December 2019, 20 samples were received, including spontaneous and non-spontaneous fermentation beers produced by homebrewers and craft brewers, as well as some samples from house cultures propagated to ferment these beers. Samples were obtained from different regions including countries such as Canada, the United States, and Israel. Furthermore, these samples had different maintenance times, ranging from a few months of storage to more than 5 years of use and propagation by the brewer. Non-spontaneous fermentation samples originated from commercial blends, bottle dregs, or cultures that were propagated and maintained by brewers, of which the exact microbial composition was not known. Spontaneous fermentation samples mostly originated through the process of exposing the beer wort to the ambient air using open fermenters. In some of these samples, it was observed that the brewers inoculated the beer using an external source that likely contained microorganisms such as wood, flowers, or fruits. Details concerning the individual samples can be found in Table 1 and Supplementary Information File S1.

In the case of beer samples, 50 mL was processed by centrifugation at $2000\times g$ for 10 min at 4 °C and the supernatant was decanted. For mixed-culture samples, a small volume ($\leq$5 mL) was resuspended in 50 mL sterile water and treated in the same manner. Then, the pelleted cells and debris were resuspended in 500 μL of 2× DNA/RNA Shield (Zymo Research (Irvine, CA, USA) and stored at −20 °C. Samples were subsequently submitted for microbiome analysis through the ZymoBIOMICS® Targeted Sequencing Service for Microbiome Analysis by the Zymo Research company.

**Table 1.** Identification and characteristics of samples of the mixed cultures received for the study.

| Sample | Origin | Material | Spontaneous Fermentation | Fruits, Woods, Flowers, or Another Microbe Source Added | Culture Maintenance Time | Commercial Strains Inoculated |
|---|---|---|---|---|---|---|
| 1 | Jerusalem (IL*) | Culture pre-pitch | YES | NO | 6–12 months | - |
| 2 | Alberta (CA) | Beer/slurry | NO | YES | 1–6 months | New W. Saison (Escarpment Labs) and Brett 'M' (Escarpment Labs) |
| 3 | Alberta (CA) | Beer | YES | YES | 6–12 months | - |
| 4 | Alberta (CA) | Beer | YES | NO | 6–12 months | - |
| 5 | Alberta (CA) | Beer | YES | YES | 6–12 months | - |
| 6 | Alberta (CA) | Beer | YES | YES | 6–12 months | - |
| 7 | Alberta (CA) | Beer | YES | YES | 6–12 months | - |
| 8 | Alberta (CA) | Beer | YES | YES | 6–12 months | - |
| 9 | Washington (US) | Culture pre-pitch | YES | NO | 2–3 years | - |
| 10 | Washington (US) | Culture pre-pitch | YES | NO | 6–12 months | - |
| 11 | Ohio (US) | Beer/slurry | NO | NO | 6–12 months | Sour Solera (Bootleg Biology); Mélange (The Yeast Bay); BugCounty (East Coast Yeast); and dregs from beer bottles |
| 12 | Nevada (US) | Beer | NO | YES | 6–12 months | Dregs from beer bottles |
| 13 | Nevada (US) | Beer | NO | YES | 6–12 months | Dregs from beer bottles |
| 14 | California (US) | Beer | NO | NO | 6–12 months | in-house culture |
| 15 | California (US) | Culture pre-pitch | YES | NO | 5–6 years | - |
| 16 | California (US) | Beer | NO | NO | 4–5 years | in-house culture |
| 17 | California (US) | Beer | YES | NO | 1–6 months | - |
| 18 | California (US) | Beer | NO | NO | 2–3 years | WY3763 (Wyeast); WY3711 (Wyeast); and WLP650 (White Labs) |
| 19 | California (US) | Beer | NO | NO | 1–2 years | WLP565 (White Labs) and dregs from beer bottles |
| 20 | Michigan (US) | Beer | YES | NO | 1–6 months | CBC-1 (Lallemand) for bottling |

* IL, Israel; CA, Canada; and US, United States.

## 2.2. DNA Extraction and Sequencing

DNA extraction was performed using a ZymoBIOMICS® -96 MagBead DNA kit (Zymo Research, Irvine, CA, USA) or ZymoBIOMICS® DNA Miniprep kit (Zymo Research, Irvine, CA) with an automated extraction platform (www.zymoresearch.com/pages/microbiome-analysis-services, accessed on 6 July 2021). Bacterial and fungal identifications were performed with 10% PhiX spike-in using next generation sequencing (NGS) on an Illumina® MiSeq™ system with a v3 reagent kit (600 cycles). First, targeted libraries were prepared for both groups. For bacteria, 16S ribosomal RNA gene-targeted sequencing was performed using a Quick-16S™ NGS Library Prep Kit (Zymo Research, Irvine, CA, USA), in which 16S primers were used to amplify the V3/V4 region of the 16S rRNA gene, maintaining good coverage and high sensitivity. Fungal internal transcribed spacer (ITS)-targeted gene sequencing was performed using the same kit described above, though ITS2 primers were used instead 16S primers, which amplifies the entire ITS region and allowes for the molecular phylogenetic sequence identification for many fungi.

PCR reactions were performed to prepare the sequencing library, controlling cycles and limiting PCR chimera formation. The quantification of final PCR products was performed with qPCR fluorescence readings and DNAs were pooled together based on equal molarity. The cleaning and quantification of final pooled libraries was performed using Select-a-Size DNA Clean & Concentrator™ (Zymo Research, Irvine, CA, USA), TapeStation® (Agilent Technologies, Santa Clara, CA, USA), and Qubit® (Thermo Fisher Scientific, Waltham, WA, USA) reagents. Negative controls (blanks) were used during all processes, as well as the ZymoBIOMICS® Microbial Community Standard (Zymo Research, Irvine, CA, USA) as a positive control, which mimics a mixed microbial community of a well-defined composition, containing Gram-negative and Gram-positive bacteria and yeasts. Additional information can be found at the ZymoBiomics™ Service website.

## 2.3. Bioinformatics Analyses

Sequencing results were used for taxonomy assignment, sample composition visualization, and alpha and beta-diversity analyses. Unique amplicon sequence variants (ASVs) were inferred from raw reads using the DADA2 pipeline [12]. The ASVs of bacteria and fungi were used to create three-dimensional principle component analysis (PCoA) plots using the matrix of paired-wise distances between samples, calculated by the Bray–Curtis dissimilarity. Uclust from Qiime v. 1.9.1 [13] was used for the taxonomy assignment using the Zymo Research Database, a 16S and ITS database that was internally designed and curated, as a reference. Results were re-analyzed by amplified sequence alignment using the nucleotide collection database from NCBI (National Center for Biotechnology Information, Rockville, MD) and the nucleotide BLAST tool. Taxonomy nomenclature and classification were analyzed using the Taxonomy Browser tool on the NCBI website (http://www.ncbi.nlm.nih.gov/Taxonomy/Browser, accessed on 8 July 2021) [14].

Sequencing results were used to separately analyze the phylogenetic relationships between groups of bacteria and fungi with MEGA v.10.1.7 software for alignment, construction, and visualization of phylogenetic trees. Sequence alignment was performed using ClustalW, which was also used to construct neighbor-joining (N-J) phylogenetic trees with 1000 bootstrap trials. Circular trees were used as templates for final figures, which had schemes and colors added using Gimp v.2.10.18 software. The microbial composition for each sample was evaluated using GraphPad Prism v.7 software, which was also used to plot the data. Samples were also analyzed for taxonomy, visualization, and interactive reporting using the Knomics-Biota system [15].

## 3. Results

Based on the different pipelines used by ZymoResearch and Knomics-Biota, with ASVs examined through the analysis of each amplified fragment using the NCBI nucleotide database, we were able to determine the metagenomes of 20 different beer samples and

starter cultures that originated from mixed fermentations made by both spontaneous and non-spontaneous methods.

### 3.1. Bacteria—16S rRNA V3-V4 Region Analysis

#### 3.1.1. Bacterial Composition

The analysis of the bacteria present in the various samples revealed 60 genera and 120 different species, with some samples containing only one species of bacteria (e.g., samples 15, 16, 17, and 18) and others containing >50 species (e.g., sample 1) (Figure 1). Based on ASV analysis, the most prevalent bacteria were *Lactobacillus acetotolerans*, which was identified in 60% of the analyzed samples (*n* = 12), followed by *Pediococcus damnosus* and *Ralstonia picketii/mannitolilytica*, both identified in 35% of the samples (*n* = 7). Although a large number of bacteria were identified at the species level, we observed that several ASVs did not allow for the differentiation of species belonging to the same genus, leading to the designation of two different species for the same ASV, such as for *L. collinoides/paracollinoids*. Only six ASVs could not be identified at the genus level, allowing only for their classification at the level of phylum, order, or family, as in the case of Myxococcales bacteria classification. The microbial composition and raw reads of each sample are available in detail in Supplementary Information File S1, while the total number of bacterial and fungal species identified is shown in Figure 2.

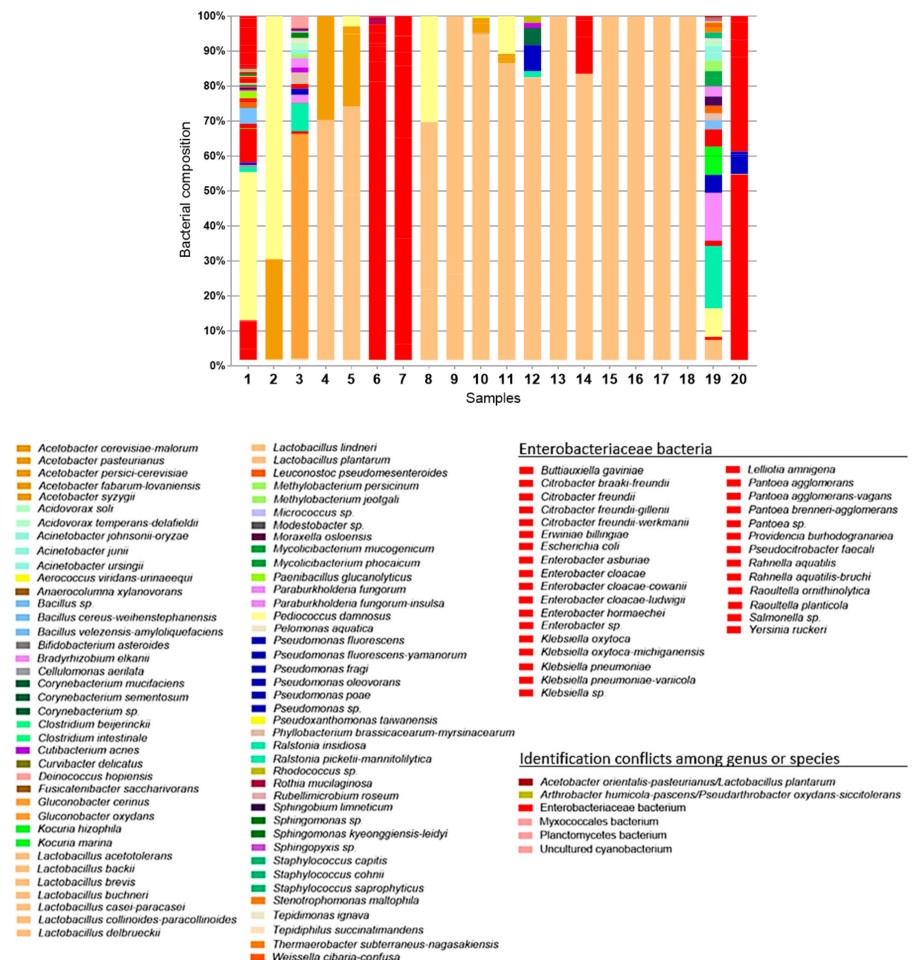

**Figure 1.** Bacterial composition of the 20 samples analyzed. Bacterial identification was performed from ASVs originating from NGS using the V3/V4 region of the 16S rRNA gene. Due to the large number of species found, colors for identification were designated according to genera, with the exception of the Enterobacteriaceae family which is identified in the graph with the red color only.

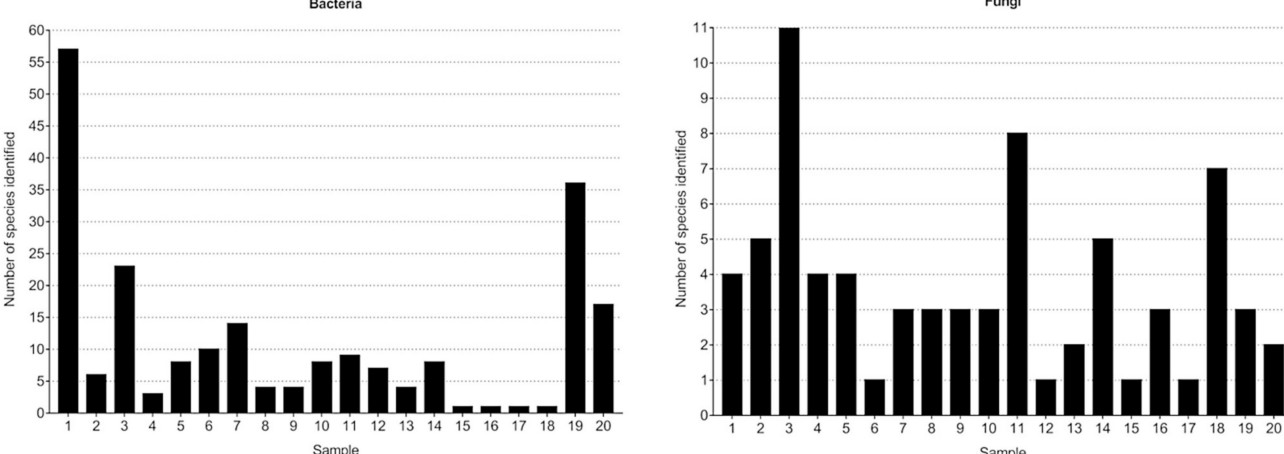

**Figure 2.** Number of species identified in each sample for bacteria (**left**) and fungi (**right**). Metagenome identification of mixed-fermentation samples made it possible to identify samples containing from just one to >55 species of bacteria, while for fungi, it was possible to detect samples with only one yeast participating in the fermentation and up to more than eight different species acting together during fermentation.

The significant presence of bacteria from the Enterobacteriaceae family was observed in different samples, which could be >15% of the ASV total composition, as in samples 1 and 14, and could even surpass 90% of the bacterial composition, as in samples 6, 7, and 20 (Figure A1, Appendix A). However, even though many of these beers were made by spontaneous fermentation and were inoculated with external sources of microorganisms, the presence of this bacterial group was <2% of the total composition of ASVs identified in most samples (*n* = 14).

Based on the differences in the microbial composition among samples, beta diversity demonstrated through PCoA plots revealed some similarities among samples according to the identified genera (Figure 3a). We highlight four main groups: group A (samples 4, 5, 8, 9, 10, 11, 12, 13, 14, 15, 16, 17, and 18), in which the genus with the highest prevalence was *Lactobacillus*; group B (samples 6, 7, and 20), in which samples were composed mainly by Enterobacteriaceae bacteria; group C (samples 1 and 2), in which *Pediococcus* (*Pediococcus damnosus* exclusively) was the genus with the highest proportion; and group D (samples 3 and 19), in which the microbial composition revealed a large presence of other ASVs detected, such as *Gluconobacter* spp. and *Ralstonia* spp. Similarities observed in the bacterial composition could also be seen in the phylogenetic tree generated through the Knomics-Biota pipeline, which grouped samples in a similar way (Figure 3b). Note that the similarity between samples 1 and 11 was due to specific ASVs of *P. damnosus*, present only in these two samples.

### 3.1.2. Phylogenetic Analysis

The construction of phylogenetic trees (Figure 4a) showed that *Lactobacillus* spp. and *Pediococcus* spp. were the main microbes from the LAB, a group of great importance in the production of sour beers. At least nine different *Lactobacillus* spp. were detected, including *L. brevis, L. plantarum, L. casei/paracasei, L. backii, L. buchneri, L. lindneri*, and *L. delbrueckii*, in addition to those mentioned above. Among the several species of *Pediococcus*, only *P. damnosus* was identified but the presence of possible subspecies could be responsible for the multiple different ASVs found. In addition, belonging to the LAB group, bacteria in the genera *Leuconostoc, Weissela*, and *Aerococcus* were likewise detected (with lower prevalence), as well as bacteria with potential probiotic activities, such as *Bifidobacterium* spp. and *Bacillus* spp. Other phylogenetically distinct groups of bacteria also demonstrated important participation in spontaneous and non-spontaneous fermentations, such as AAB *Gluconobacter* spp. and *Acetobacter* spp., and especially *G. oxydans* and *A. pasteurianus*.

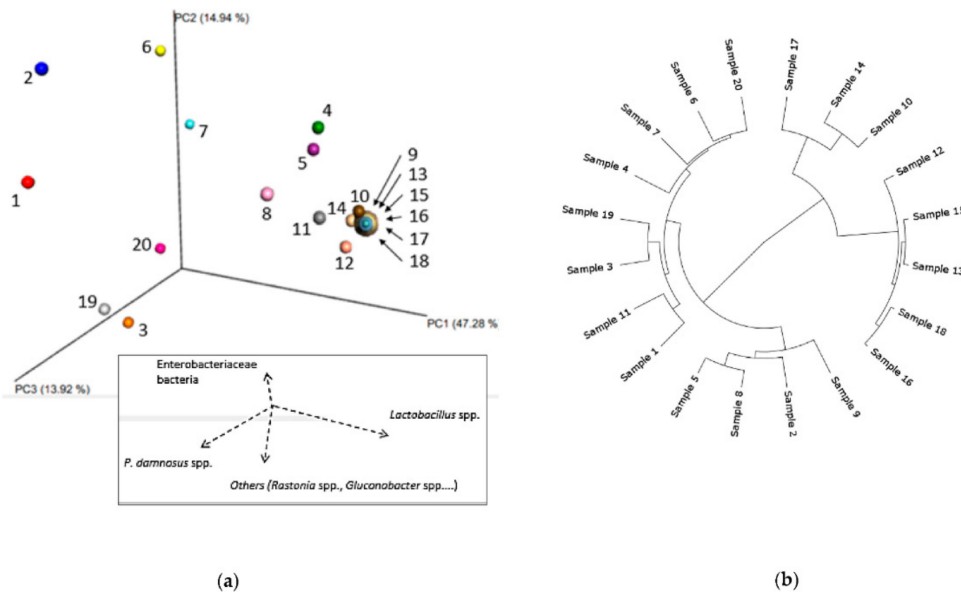

**Figure 3.** Sample similarity based on bacterial composition. (**a**) PCoA tridimensional plot created using the matrix of pairwise distances between samples calculated by the Bray–Curtis dissimilarity using genera found in the microbial composition. 3D images, schematic representation, and visualization were generated using the EMPeror tool. Below the PCoA plot is a schematic representation based on the sample composition, in which arrows show different directions for locating the samples on the graph according to the presence and concentration of Enterobacteriaceae bacteria, *Lactobacillus* spp., *P. damnosus*, and other bacteria such as *Ralstonia* spp. and *Gluconobacter* spp. (**b**) Phylogenetic tree showing clustering of the samples by similarity of their taxonomic composition, calculated by the Bray–Curtis dissimilarity at the ASV level using Ward's method.

Several species of the genus *Pseudomonas* were also detected, mainly in sample 20, in which it was possible to identify eight different ASVs related to this genus, corresponding mainly to *P. fluorescens*. Although *Pseudomonas* spp. are mainly known because of the pathogenic bacterium *P. aeruginosa*, this organism was not detected in the studied samples. A special emphasis was given to the Enterobacteriaceae family in the phylogenetic tree as they corresponded to a large number of identified ASVs and thus represented a vast number of species found at different concentrations. The samples' metagenomes showed the presence of a large number of genera included in this family, among them *Enterobacter* spp., *Salmonella* spp., *Klebsiella* spp., and *Pantoea* spp. During the identification of the ASVs corresponding to these bacteria, it was possible to observe a high similarity among the genera, which made specific identification difficult in many cases, especially at the species level.

In analyzing the metagenomes discovered (Figure 4b), three main phyla were identified with high prevalence: Proteobacteria (66.67% of the identified bacteria belonged to this phylum), Firmicutes (19.38%), and Actinobacteria (11.63%). ASVs from other phyla were also detected, though at smaller proportions: Deinococcus-Thermus (0.78%), Cyanobacteria (0.78%), and Planctomycetes (0.78%). Almost half of the identified bacteria belonged to the Gammaproteobacteria class, which includes families such as Enterobacteriaceae, Pseudomonadaceae, and Acetobacteriaceae, among others, totaling to 17 different bacterial families. Though most studies and applications involving mixed-fermentation beers focus on the Lactobacillaceae family, it was related to only 7.75% of ASVs, representing a small portion of the variety of microorganisms identified.

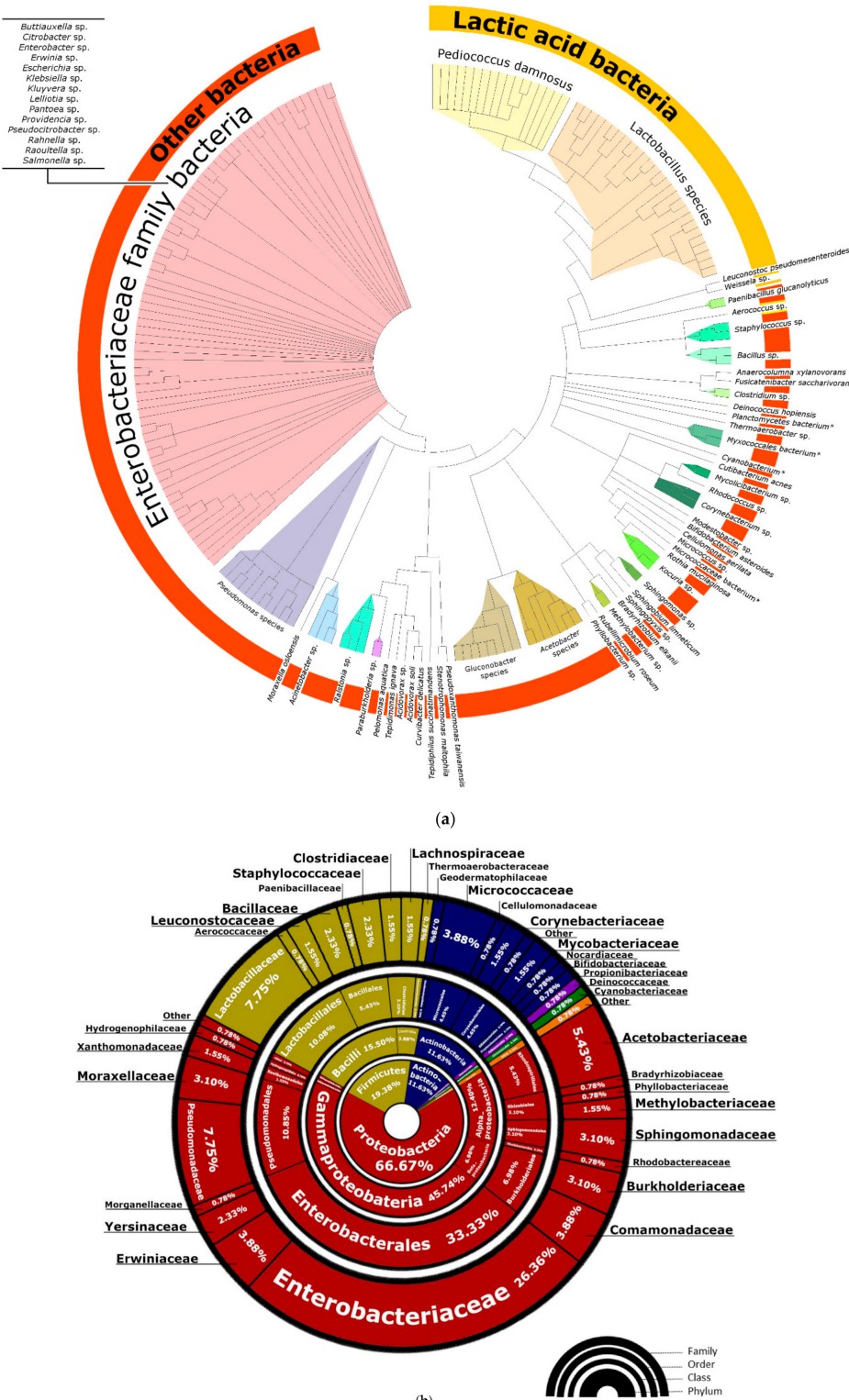

**Figure 4.** Phylogenetic analysis and abundance of bacterial taxa. (**a**) Phylogenetic tree constructed based on the ASVs of 120 bacterial species found in the mixed-culture samples. Phylogenetic relationships were made using MEGA v.10.1.7 software for alignment, construction, and visualization of the phylogenetic tree. Sequence alignment was performed using ClustalW. Supplementary Information File S2 contains a high-resolution version of Figure 4a, suitable for zooming and enlargement. (**b**) The image shows the prevalence of taxonomic classifications according to the genera and species found in the bacterial microbiomes. Higher taxonomic classifications are shown in circles near the image center, while lower classifications are shown toward the outer edge. Images were constructed using MEGA and GIMP 2.1 software.

*3.2. Fungi—ITS2 Region Analysis*

3.2.1. Fungal Composition

NGS targeting the ITS2 region was able to detect 19 genera and 26 different species of fungi in the 20 analyzed samples (Figure 2). Among them, yeasts were the most identified microorganisms, while ASVs from filamentous fungi, molds, and more complex fungi (e.g., mushrooms) were also identified in smaller proportions. Samples were basically dominated by two main yeasts: *Brettanomyces bruxellensis* and *S. cerevisiae*, which were present in 75% (*n* = 15) and 65% (*n* = 13) of the samples, respectively (Figure 5). Microbial composition analyses revealed that in the vast majority samples (*n* = 16), fermentation was carried out by at least two different species of fungi. Of these two, an association between *Brettanomyces* spp. and *Saccharomyces* spp. was shown to be recurrent (*n* = 7), in which both exceeded at least 5% concentration in the ASVs found in each sample.

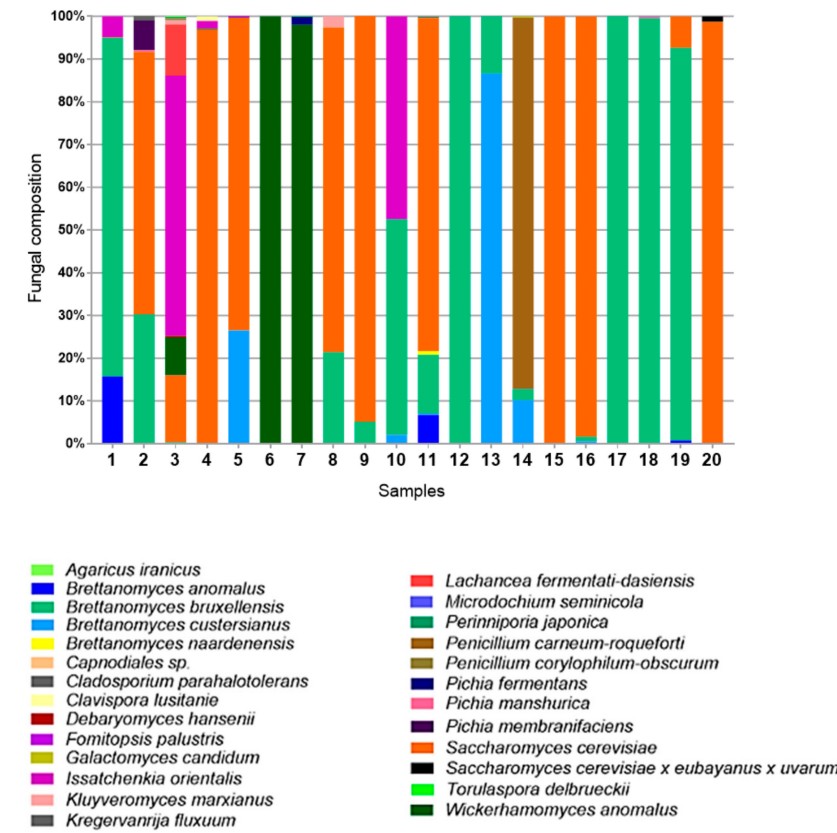

**Figure 5.** Fungal composition of the 20 samples analyzed. Fungal identification was performed from ASVs originating from NGS of the ITS2 region, amplifying the entire ITS region. Colors for identification were designed according to the fungal species as shown below the plot.

Other yeasts such as *Issatchenkia orientalis* (also known as *Pichia kludriavzevii*, *Candida krusei*, and *Candida glycerinogenes*) and *Wickerhamomyces anomalus* (or *Pichia anomala*) also participated in these mixed-fermentation beers, which could be detected in 30% (*n* = 6) and 25% (*n* = 5) of the samples, respectively, with a large presence in samples 3, 6, 7, and 10. It is interesting to note that not only were yeasts identified but also more complex fungi, such as *Penicillium* spp. which accounted for most of the ASVs, were found in sample 14. Overall, we observed that several fungal species were often present in mixed-culture samples, with more than seven species found in samples 11 and 18, and over 11 different fungi detected in sample 3.

The beta diversity of the cultures was evaluated according to the genera identified in the microbial composition of each sample. In these analyses, we could verify the distribution of samples into four groups (Figure 6a) by similarities between identified

microorganisms: group A (samples 4, 9, 15, 16, and 20), in which >95% of the sample was composed of *S. cerevisiae*; group B (samples 1, 12, 13, 17, 18, and 19), in which >90% of the identified ASVs corresponded to *Brettanomyces* spp.; group C (samples 2, 5, 8, and 11), in which approximately 70% of the sample was represented by *S. cerevisiae*, 25% by *Brettanomyces* spp., and 5% by other fungi; and group D (samples 3, 6, 7, 10, and 14), in which other genera such as *Wickerhamomyces*, *Issatchenkia*, *Penicillium*, and *Lachancea* dominated.

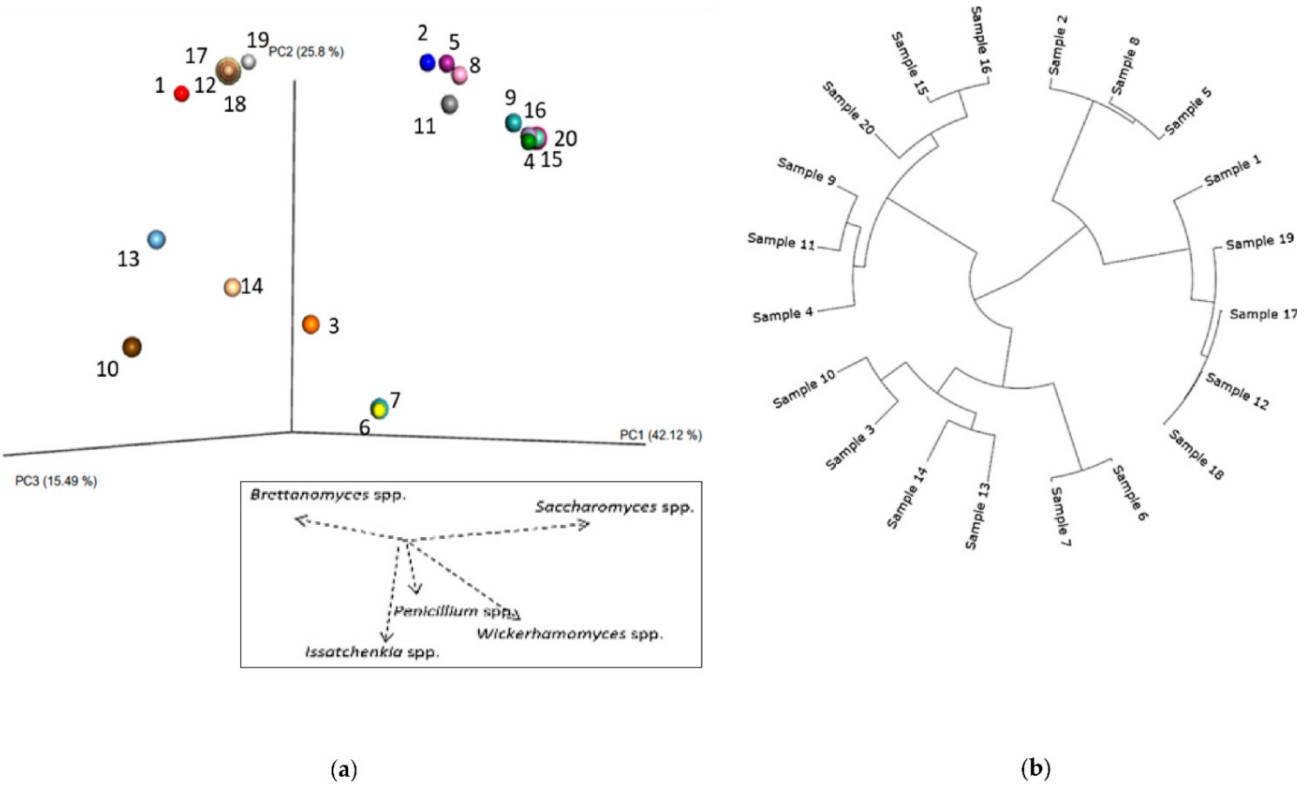

(**a**)                    (**b**)

**Figure 6.** Sample similarity based on fungal composition. (**a**) PCoA tridimensional plot created using the matrix of pairwise distances between samples calculated by the Bray–Curtis dissimilarity using the genera found in the microbial composition. Below the PCoA plot is a schematic representation based on the sample composition, in which arrows demonstrate different directions for sample localization according to the presence and concentration of *Brettanomyces* spp., *Saccharomyces* spp., *Penicillium* spp., *Issatchenkia* spp., and *Wickerhamomyces* spp. (**b**) The tree shows clustering of the samples by similarity of their taxonomic composition, calculated by the Bray–Curtis dissimilarity at the ASV level using Ward's method.

The tree constructed based on the hierarchical clustering between ASVs detected for ITS2 (Figure 6b) reveals that samples tended to follow the same relationship observed in Figure 6a, though some samples such as 13 and 14 had greater hierarchical approximation because similar ASVs referring to *Brettanomyces custersianus* were found in significant proportions (86% and 10% of the ASVs identified in these samples, respectively). Similarly, samples 9 and 11, though organized in different groups in Figure 6a, presented a phylogenetic relationship in this analysis because there were specific ASVs corresponding to *B. bruxellensis* and *Brettanomyces anomalus* found exclusively in their microbial composition. The other samples displayed phylogenetic relationships in accordance with those observed in the PCoA analysis, confirming similarities in the fungi composition among the samples evaluated in this study.

### 3.2.2. Phylogenetic Analysis

The construction of a phylogenetic tree using the fungal species detected in this study reveals different subspecies of both *B. bruxellensis* and *S. cerevisiae* (Figure 7a). Their

excellent growth and adaptability in beer wort favors the presence of these yeasts, as well as the presence of other *Brettanomyces* spp. such as *B. anomalus, B. naardenensis*, and *B. custersianus*, all of which were identified in the spontaneous and non-spontaneous fermentation beers. In addition, multiple species from the *Pichia* genus were detected, such as *P. fermentans* and *P. membranifaciens*, with an emphasis on four subspecies of *I. orientalis* that were identified as a large proportion in some samples. We observed that different ASVs referring to the fungus *Penicillium* spp. were present in these samples, as well as more complex fungi such as *Perenniporia japonica* and *Fomitopsis palustres* (ASVs identified only in samples 3 and 18, respectively), whose presence is generally related to wood colonization.

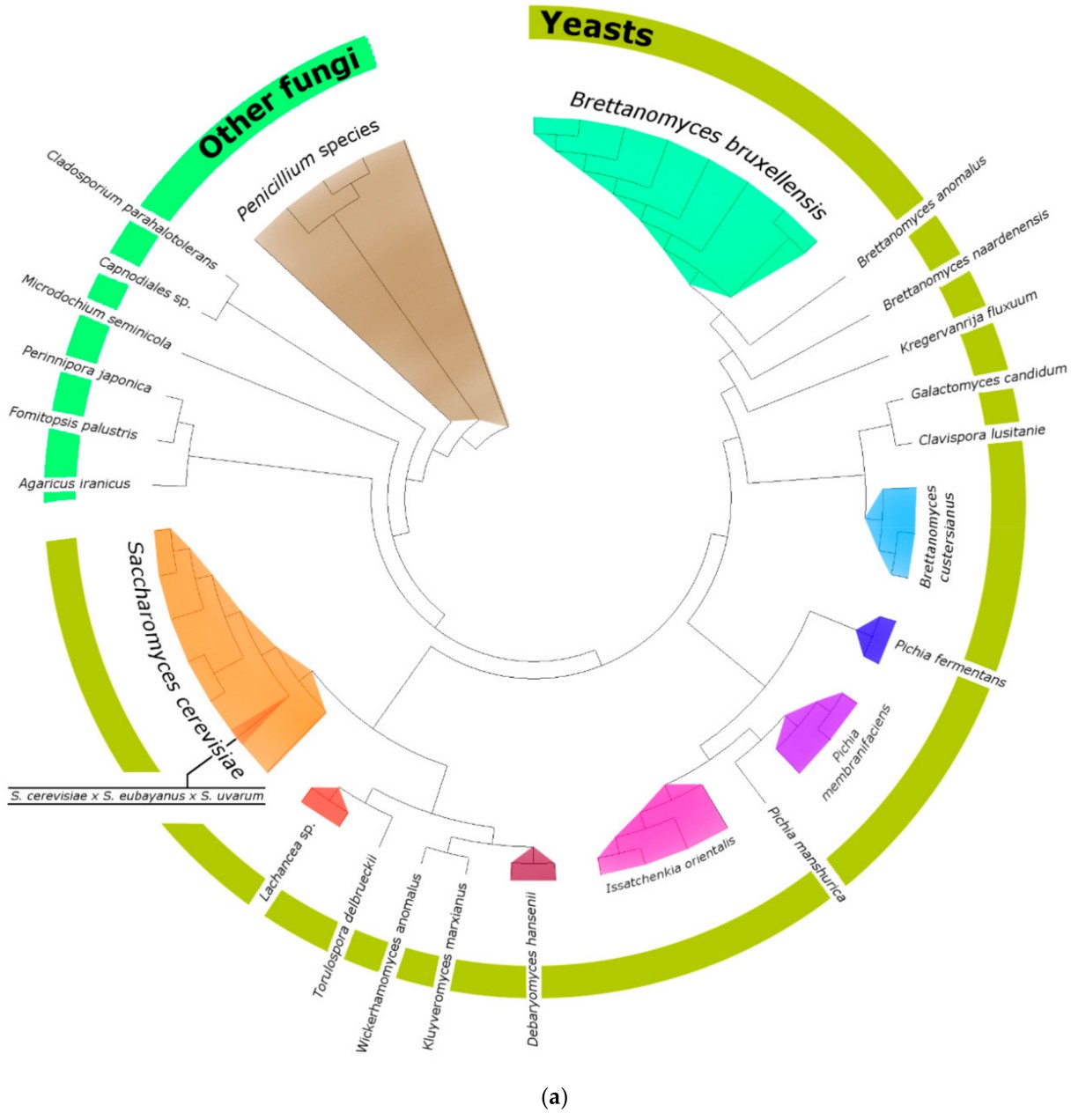

(**a**)

**Figure 7.** *Cont.*

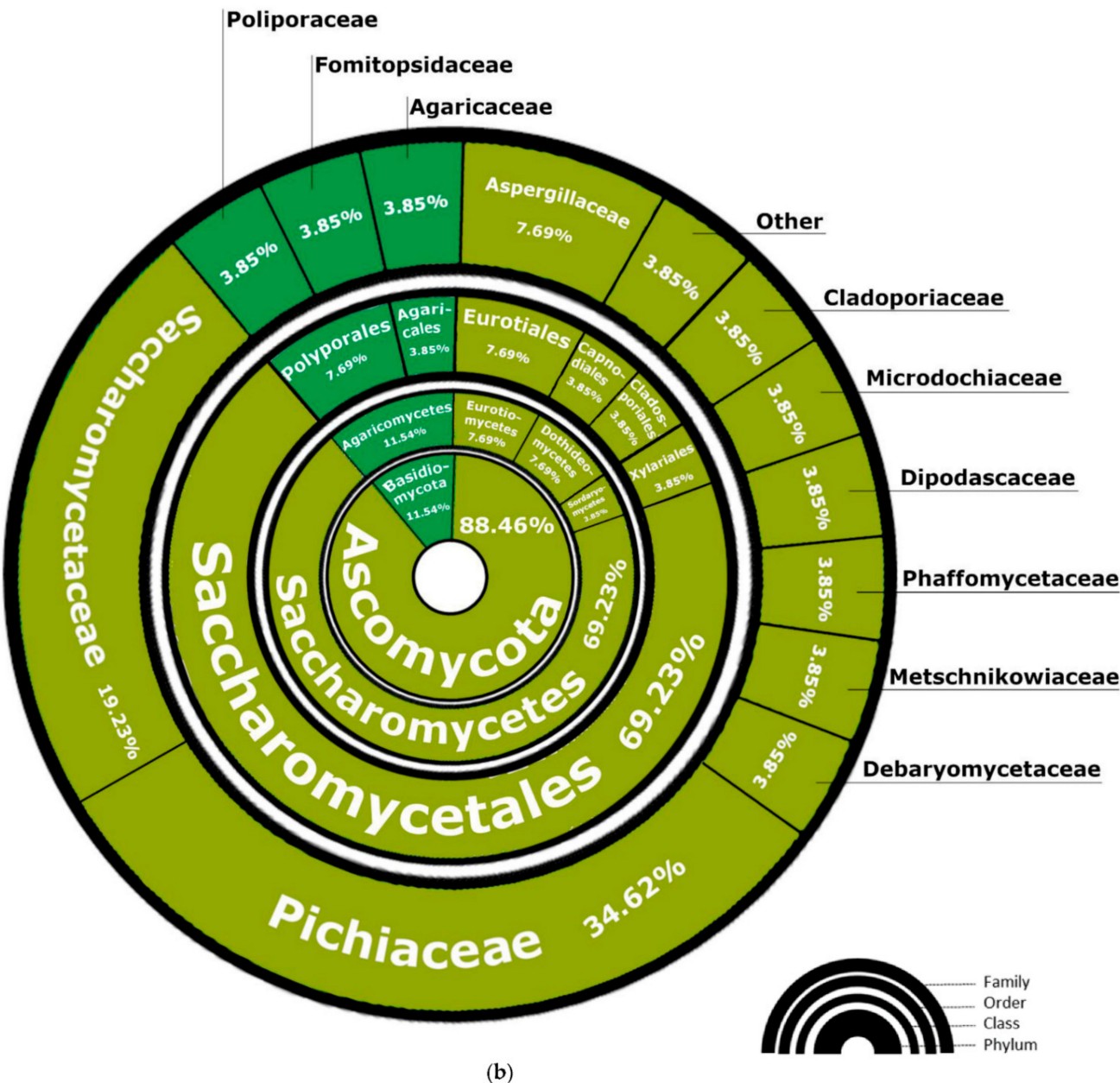

**Figure 7.** Phylogenetic analysis and abundance of fungal taxa. (**a**) Phylogenetic tree constructed based on the ASVs of 26 fungal species found in the mixed-fermentation beer samples. Supplementary Information File S3 contains a high-resolution version of Figure 7a, suitable for zooming and enlargement. (**b**) The image shows the prevalence of taxonomic classifications according to genera and species found in the fungal microbiomes. Higher taxonomic classifications are shown in circles near the image center, while lower classifications are shown toward the outer edge.

Fungi from the Ascomycota phylum, which contains yeasts, accounted for 88.46% of the ASVs found, as well as the Basidiomycota phylum to a lesser extent (11.54%) (Figure 7b). The Pichiaceae family, which contains yeasts of the genera *Brettanomyces*, *Pichia*, *Kregervanrija*, and *Issatchenkia*, displayed the highest proportion among the families classified (34.62%) in this study. The *Saccharomyces*, *Torulaspora*, *Lachancea*, and *Kluyveromyces* genera belong to the Saccharomycetaceae family, a taxonomic classification that comprises genera of great importance in mixed-fermentation beer production, and such microbes were responsible for 19.23% of the fungi ASVs classified. Other families were responsible for the classification of only one genus, representing 3.85% each of the total yeasts found.

## 4. Discussion

### 4.1. Exploring the Microbiomes of Mixed-Fermentation Beers

Elucidating the microbiomes of mixed-fermentation beers is important to understand the participation of microorganisms in this fermentation niche. The production of sour beers from spontaneous fermentations is associated with inconsistencies in product quality, unpredictability in fermentation results, and extra time to consumption [8], as it is often not known which microorganisms will act in the beer wort exposed to the thousands of bacteria and fungi present in the environment or after the repitching a slurry from previous mixed-fermentation batches. Even brewers who work with spontaneously fermented beers for years or decades very often do not know which wild microbes are fermenting their beers, thus it is important to reveal the microbiomes of mixed cultures responsible for fermenting normal beer wort into high-value sour beers. Thus, identifying and characterizing the metagenomes present in mixed fermentation beers and mixed cultures samples allows one to predict which microorganisms typically participate in these fermentations and formulate new bacterial and yeast blends to mimic this fermentation process in a controlled and reproducible way.

Preliminary studies performed by our research group concerning the metagenomes of commercial beers made by mixed fermentation and the barrels used in their maturation corroborate the data presented here, where *L. acetotolerans*, *L. brevis*, *L. buchneri*, *B. bruxellensis*, and *S. cerevisiae* were commonly observed in the samples (data not shown). Though different microorganisms have been detected at lower concentrations, the metagenome data in this study are in accordance with observations by other researchers, such as Bokulich et al. (2012) [10], Bokulich et al. (2015) [16], De Roos et al. (2019) [3], and Tyakht et al. (2021) [15]. We observed in our work that dozens of bacterial species can be found through NGS using the V3/V4 region of the 16S rRNA, revealing that beers produced spontaneously and non-spontaneously represent an interesting source for the identification of new microorganisms that cooperate or compete amongst themselves in the consumption of substrates present in beer wort [17]. Their presence is important in mixed-fermentation beers as they are able to enhance the production of specific flavors and acidify the wort through lactic acid production, which are typical characteristics in these beer styles [2].

Fungal ASV detection by sequencing the ITS2 region revealed microbiomes based on two main genera: *Brettanomyces* spp. and *Saccharomyces* spp., which were found in the vast majority of samples corresponding to >90% of the ASVs identified per sample. However, there is still a little-known world of unconventional yeasts to be discovered and characterized, which is comprised of less common genera such as *Issatchenkia* and *Wickerhamomyces*. The fungal microbiomes presented a lower variability in the number of species when compared to bacteria, which may be related to the avidity in glucose consumption by commercial strains (reducing the concentration of easily assimilated carbohydrates in the medium) and the competition among yeasts, which can present killer characteristics, producing secondary metabolites that aim to stop the multiplication of cross-feed competitors [18,19]. Even so, 12 different yeast genera were detected as participating in these fermentations.

### 4.2. Lactobacillus spp. and Pediococcus spp. Are Often Identified in Mixed-Fermentation Beers

In recent years, the application of LAB has been explored in sour beer production, with emphasis on bacteria in the *Lactobacillus* genus as not just contaminants in beer but as interesting tools for acidification and the production of new flavors [20]. Although the vast majority of these bacteria are not tolerant to hop alpha-acid concentrations of up to 20 international bitterness units (IBUs), some species and subspecies adapt to the adverse conditions that exist in beers, becoming tolerant to this and others selective pressures such as hydrostatic pressure, alcohol, and low pH [21]. In our study, we observed nine *Lactobacillus* spp., including *L. acetotolerans* (with eight identified subspecies), *L. backii*, and *L. plantarum,* a genus present in 18 of the 20 samples analyzed. Several authors highlight the presence of *Lactobacillus* spp. in the microbiomes of mixed-fermentation beers, such

as Tyakht et al. (2021) [15] and Spitaels et al. (2014) [7], who detected these bacteria in American Coolship Ales, Wild Ales, Sour Ales, and Belgian Lambic Beers.

*P. damnosus* was the only identified species from the *Pediococcus* genus but 18 subspecies were detected in this study. Similar to *Lactobacilllus* spp., *Pediococcus* spp. are commonly considered as beer and manufacturing environmental contaminants, a characteristic that encourages research into the identification of resistance factors mainly acquired through specific plasmids and genes [17,22]. In Lambic-style beers (produced by spontaneous fermentation), *P. damnosus* is one of the main isolated species and can be easily identified as part of the microbiome of the interior surface of wooden barrels [1], in the air, and on other brewery equipment surfaces [23] mainly due to its high oxidative stress resistance and hop tolerance [4]. In our study, ASVs corresponding to these bacteria were found in seven samples, representing up to 70% of the bacterial microbiome, as in sample 2.

Biological acidification during the brewing process by the action *Lactobacillus* spp. or *Pediococcus* spp. has several benefits: flavor stability, greater zinc bioavailability, fast final attenuation, lower wort viscosity, and smoother hop bitterness, among others [2]. In pre-fermentation tests using these bacteria, impacts are observed on volatile compound and organic acid production, which result in significant differences in the sensorial characteristics of sour beers [20]. These bacteria can also use carbohydrates that are not metabolized by conventional yeasts, such as maltotriose, maltotetraose, and cellobiose, resulting in beer over-attenuation [8,24]. Although *Pediococcus* spp. are historically reported as responsible for high levels of diacetyl production and causing viscous "sick beers" (through exopolysaccharide secretion), these bacteria have been intentionally combined with *Brettanomyces* spp. to generate a deeper acidity and mouthfeel in various mixed-fermentation beers [24]. Some *Brettanomyces* strains have β-glucosidase enzyme activity, permitting the yeast to degrade the exopolysaccharides produced by LAB and demonstrating the importance of microbial consortia found in mixed-fermentation beers [4].

### 4.3. Bacteria of the Enterobacteriaceae Family and Their Presence in Spontaneous and Non-Spontaneous Fermentations

Enterobacteriaceae bacteria are well-known contaminants in spontaneously fermented beers, participating in the initial stages of fermentation (also known as the Enterobacteriaceae phase) that starts on day one and can last approximately 1 month [1]. In our work, ASVs corresponding to different bacteria in this family were identified, such as *Rahnella* spp., *Klebsiella* spp., *Enterobacter* spp., *Providencia* spp., *Escherichia* spp., *Erwiniae* spp., and *Pantoea* spp. In samples 6, 7, and 20, we found that almost the entire bacterial microbiome corresponded to Enterobacteriaceae, with emphasis on the species *R. aquatilis*, *K. oxytoca*, *E. asburiae*, and *E. billingiae*, which were present at levels of >20% of the ASVs found in these samples. Using methods for the isolation of microorganisms in specific-culture media, Spitaels et al. (2014) [7] could verify that after 2 months of maturation, only *P. damnosus* was found in beers in which Enterobacteriaceae were previously recovered during the initial fermentation period. Thus, it is noteworthy that although we detected ASVs corresponding to these bacteria in the samples in our study, this does not imply that these microbes were viable at the time of sample collection.

Different studies have successfully isolated and identified Enterobacteriaceae in spontaneously fermented beers, whether American Coolship Ales, Belgian Lambics, or Wild Ales [7,9,15]. In such cases, these bacteria were highly abundant (>8% of ASVs) in six beer samples. Although they are responsible for off-flavor production and can be harmful to health through biogenic amine production [1,9], these bacteria are also linked to the production of specific flavors related to young Lambics, such as 2,3-butanediol, ethyl acetate, higher alcohols, acetic acid, lactic acid, succinic acid, and fatty acids [10,24]. Manual wort acidification prior to fermentation is a technique that may lower the Enterobacteriaceae concentration during early fermentation and is often used by lambic brewers to shorten the Enterobacteriaceae phase [4].

### 4.4. Other Bacteria

The metagenomes of our mixed-culture samples were not restricted to bacteria from the Enterobacteriaceae and Lactobacillaceae families. Several different species were often present, mainly AAB from the genera *Gluconobacter* and *Acetobacter*. Indeed, *G. oxydans* corresponded to 64.23% of the ASVs found in sample 3 and different *Acetobacter* spp. were present in concentrations of >20% in samples 2, 4, and 5. Their presence is likely due to the fact that they are able to tolerate hop alpha-acids and ethanol concentrations up to 10%, as well as being responsible for ethanol oxidation with the formation of organic acids, mainly acetic acid (one of the flavors commonly associated with spontaneously fermented beers) [25].

Similar to our findings, Tyakht et al. (2021) detected the presence of the Leuconostocaceae and Acetobactereaceae families at abundant levels (>5%) in the microbiomes of two wild beers. Not only are LAB important in the flavor bouquet construction of sour beers, but other less conventional bacteria can also be highlighted, such as *Acetobacter* spp. which are related to the production of beers with high contents of 5-methyl-furfural, flavonoids, and 2- and 3-methyl butanol [9]. When associated with the presence of lactic acid, acetic acid addition by these bacteria (at adequate amounts) can increase the sensory complexity of the beer, resulting in the construction of the "layered flavors" so important in traditional Belgian sour beers [4,24]. However, excessive production of acetic acid and acetoin by AAB is normally avoided through the use of full and well-sealed wooden casks. This maintains a yeast pellicle at the wort/air surface that enables microaerobic conditions, consequently limiting AAB growth [4].

Bacteria of the genus *Pseudomonas* were also detected. Even though a large variability in the of ASVs for *Pseudomonas* spp. was found only in sample 20, *P. fluorescens* was identified in 20% of the studied samples (*n* = 4), with concentrations ranging from 0.3–8% in samples in which it was identified (samples 1, 3, 12, and 19). This genus was also identified by Rodhouse (2017) [26], mainly in malt samples, with a decrease in its detection during the brewing process (mashing, boiling, and bottling). Considering spontaneously fermented beers are usually exposed to the environment, it is possible that these bacteria are transported from the raw brewing ingredients (mainly malt) to the wort during the open-air fermentation process [27].

Aside from *Pseudomonas* spp., other Gram-negative bacteria found in significant concentrations were *Ralstonia* spp., representing 17% of bacterial ASVs found in sample 19 and 8% in sample 3. Both genera were also detected at small concentrations (<$10^3$ cell/mL) by other researchers such as Takahashi et al. (2015) [28] who observed the presence of these bacteria in the early and intermediate stages of beer-like beverage fermentation and Bokulich et al. (2012) [10] who detected them in American Coolship Ales. Other bacteria such as *Acinetobacter* spp., *Stenotrophomonas* spp., *Sphingomonas* spp., and *Staphylococcus* spp. were also detected in small amounts in the metagenomes of mixed-fermentation beers by Bossaert et al. (2021) [9]. However, De Roos et al. (2019) [3] detected a high relative abundance of the same bacteria on the surfaces of wooden barrels used in beer fermentation.

### 4.5. Fungi besides the Brettanomyces spp. and Saccharomyces spp.

In addition to traditional and well-known *Brettanomyces* spp. and *Saccharomyces* spp., other yeasts were present in the microbiomes of our mixed-culture samples. Genera such as *Debaryomyces*, *Lachancea*, and *Pichia*, and mainly *Issatchenkia* and *Wickerhamomyces*, were identified as part of the samples' fungal microbiomes. *I. orientalis*, a yeast known for its industrial application in bioethanol production and participation in the construction of the aroma bouquet in wines [29], has been used in co-fermentations with *S. cerevisiae* for the production of beers with higher levels of fruity esters [30]. In our study, we identified this yeast at high concentrations (>45% of the ASVs found) in the fungal microbiomes of samples 3 and 14, demonstrating its adaptability to beer wort and its participation in the fermentation process.

*Wickerhamomyces anomalus* and *D. hansenii* are yeasts very often detected in the air surrounding coolships, as well as observed participating in Lambic beer fermentation [4]. The use of *W. anomalus* has already been studied in a controlled way in the production of beers, classifying this yeast as a potential organism for the primary souring technique with contribution to fruity organoleptic profiles [31,32]. *W. anomalus* was detected in our study, comprising almost the entire fungal microbiome (>98%) in samples 6 and 7, as well as being relatively abundant in sample 3 (9% of the ASVs). Although Tyakht et al. (2021) [15] did not detect the presence of *W. anomalus* in the beer samples analyzed, Spitaels et al. (2014) [7] confirmed its participation in Lambic beer fermentation, identifying relevant levels (>20%) in the microbiomes of beers after 24 months and detecting its presence both in the air of the brewery environment and on the external surface of casks.

Varied *Pichia* spp. such as *P. manshurica* and *P. fermentans* were also found in the analyzed samples, generally at concentrations of <2%. However, *P. membranifaciens* was the only species of this genus that could be identified above this rate in the microbiome of a sample, representing 7% of the ASVs found in sample 2. Although in our study the concentration of *Pichia* spp. was considered low, other studies such as Bossaert et al. (2021) [9] identified *P. membranifaciens* as comprising up to 98% of the OTUs (operational taxonomic units) found in beers at specific maturation periods and also revealing that beer maturation over time leads to changes in the microbiome. *Lachancea* spp. are also reported to participate in mixed-fermentation beers, mainly being responsible for pH decrease in sour beers through lactic acid production [31,33]. *L. fermentati/dasiensis* was detected in our work in sample 3, with a relatively abundant concentration of 12% of the fungal microbiome.

The beer microbiomes in our study also revealed that among the most abundant families of fungi, Aspergillaceae represented around 7% of the identified ASVs. Cason et al. (2020) [34] observed a similar relative abundance (6.7% of OTUs) in Sesotho beer samples, a traditional South African style produced by spontaneous fermentation. Fungi of the *Penicilllium* genus, which is part of this family, are related to gushing and their presence has been detected inside beer and wine barrels, mainly participating in biofilm formation on porous surfaces [3,35]. These fungi are capable of producing enzymes responsible for the degradation of lignocellulose and beta-glycans, contributing to changes in beer color and aroma [36]. Other researchers such as Bossaert et al. (2021) [9] and De Roos et al. (2019) [3] have also identified *Penicillium* spp. in their samples. However, we found only one sample (sample 14) with >80% of fungal ASVs related to *Penicillium carneum/roqueforti*. ASVs referring to fungi with advanced and more complex structures (e.g., *P. japonica*) were also detected in samples at levels between 0.05% and 0.5%, indicating little relevance in the microbiomes of the analyzed beers.

### 4.6. The Importance of Traditional Yeasts in Mixed-Fermentation Beers

*Brettanomyces* (also known as *Dekkera*), perhaps the second most important genus of yeast in mixed-fermentation beers, can be found in large proportions in spontaneously and non-spontaneously fermented beers, associated with the production of phenolic and esterified volatile compounds [8]. The ability of *Brettanomyces* ssp. to metabolize complex sugars, produce acetic acid, generate a characteristic aromatic profile, and cause "super-attenuation" in beer wort has aroused interest in their use [32,37], which has led to several production laboratories around the world distributing these yeasts commercially.

Here, *B. bruxellensis* was the species with highest proportions found in the ASVs of this genus, totaling to >90% of the fungal microbiome in some samples such as 12, 17, and 19. Sobel et al. (2017) [38] also highlighted the large presence of *B. bruxellensis* in traditional beers from countries such as Belgium, Italy, and Switzerland, detecting it in the metagenomes of 36 samples (*n* = 39). Not only did we find *B. bruxellensis* in large proportions in our samples but also *B. anomalus* (15% of the ASVs in sample 1) and *B. custersianus*, which comprised >85% of the ASVs in sample 13. These species present differences in sugar metabolism, aggregation, and flavor production, leading to interest

in their use during beer fermentation [39]. It is noteworthy that the ASVs corresponding to *Brettanomyces* spp. and *Saccharomyces* spp. found in samples 2, 11, and 18 are possibly not related to wild strains but rather to the use of commercial blends inoculated at the beginning of fermentation.

The identification of the yeasts in the metagenomes of our samples revealed only one species in the *Saccharomyces* genus, *S. cerevisiae*, which together with *S. pasteurianus* are the brewers' yeasts most used for beer fermentation and are widely commercially available through propagation laboratories [40]. Wild strains of *S. cerevisiae* can offer interesting characteristics such as the extracellular production of different secondary metabolites (related to the production of new aromas) and tolerance to different stress conditions (e.g., salinity, temperature, and high levels of ethanol) [19]. These strains have mainly been isolated and characterized from spontaneous fermentations of traditional fermented beverages such as Kveik strains, isolated from Norwegian Kveik Beer [41]. In seven samples, in which there was no commercial *S. cerevisiae* inoculum, ASVs corresponding to *S. cerevisiae* were detected, which may be related to the presence of wild strains acting during fermentation. However, because the detection of this yeast may be related to the cross-contamination of utensils and fermenters by the presence of commercial *S. cerevisiae* on such equipment [16], further studies must be conducted to confirm these ASVs as originating from wild strains.

Among the eight strains of *S. cerevisiae* detected in the metagenomes here, only one was identified as a hybrid of this yeast with another species (*S. cerevisiae* × *S. eubayanus* × *S. uvarum*). It was found at a low concentration (<2% of the ASVs) in just one sample (#20), a beer produced through spontaneous fermentation. Yeasts with hybrid genotypes are generally reported in beer and wine, where their presence is caused by interspecific hybridization during the diversification and adaptation of yeasts to the industrial niche [42].

## 5. Conclusions

The microbiome analysis of 20 samples of mixed cultures with different origins allowed us to deepen our knowledge of the metagenomes of the cultures used for beer production, identifying the distribution and concentration of bacteria and fungi in these samples. Exploring new microorganisms and their strains that adapt to and ferment beer wort is an important key factor in the rational development of new blends for brewers to ensure that the production of various beers and their flavors become reliable and reproducible. Based on these data, we conclude that this is a vast field that needs to be further explored, with potential for industrial applications and in the development of basic and applied science. Future work connecting the bacterial compositions of mixed cultures used for fermentation to metadata on the base wort, kinetics of fermentation, and organoleptic compounds produced in various sour beers will enable brewers to construct bespoke mixed cultures the generate desired sensorial results.

**Supplementary Materials:** The following are available online at https://www.mdpi.com/article/10.3390/fermentation7030174/s1, Supplementary Information S1: Raw reads, ASVs sequences, microorganisms' identification, and prevalence in each sample (Microsoft Excel file).

**Author Contributions:** Conceptualization, R.E.A.P. and M.L.B.; methodology, R.E.A.P. and M.L.B.; software, R.E.A.P., F.P.L.L. and M.L.B.; validation, R.E.A.P., F.P.L.L. and M.L.B.; formal analysis, R.E.A.P., F.P.L.L. and M.L.B.; investigation, R.E.A.P. and M.L.B.; resources, R.E.A.P., F.P.L.L. and M.L.B.; data curation, R.E.A.P. and M.L.B.; writing—original draft preparation, R.E.A.P.; writing— review and editing, R.E.A.P., F.P.L.L. and M.L.B.; visualization, R.E.A.P.; supervision, F.P.L.L. and M.L.B.; project administration, F.P.L.L. and M.L.B.; funding acquisition, R.E.A.P., F.P.L.L. and M.L.B. All authors have read and agreed to the published version of the manuscript.

**Funding:** This research was funded by donors to the Experiment.com project "Mixed culture metagenomics of the microbes making sour beer" (DOI 10.18258/13495).

**Data Availability Statement:** The data presented in this study are available in the Supplementary Information.

**Conflicts of Interest:** The authors declare no conflict of interest.

## Appendix A

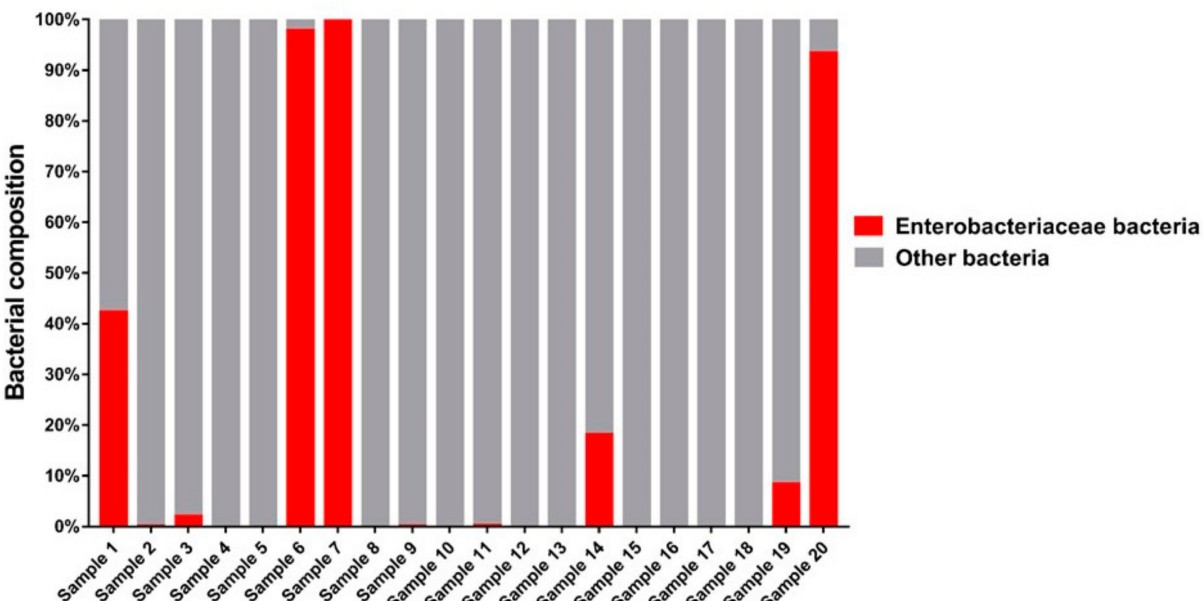

**Figure A1.** Enterobacteriaceae family presence in the bacterial composition of the analyzed samples. Bacteria from the Enterobacteriaceae family are commonly detected in the first fermentation phase of mixed-fermentation beers produced by spontaneous fermentation (De Roos and De Vuyst, 2019) and in our study they could be identified in relevant proportions in samples 1, 6, 7, 14, 19, and 20 (>8% of the ASVs detected in each sample).

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
