# Peer review of "Mixed-Culture Metagenomics of the Microbes Making Sour Beer"

_fermentation, doi:10.3390/fermentation7030174_

Round 1

Reviewer 1 Report

This manuscript "Mixed culture metagenomics of the microbes making sour beer” is about the microbial communities in several beer products. However, there are some serious question.

Major comments: 

  1. (Spontaneous vs non-spontaneous) Seven samples were produced by non-spontaneous fermentation. However, it was lack for information about non-spontaneous fermentation. For example, starter information and relationship with current results.

  1. (Beer conditions) Actually, microbial communities affected by external factors such as pH, and alcohol contents. Therefore, it need to revise.

Author Response

  1. (Spontaneous vs non-spontaneous) Seven samples were produced by non-spontaneous fermentation. However, it was lack for information about non-spontaneous fermentation. For example, starter information and relationship with current results.

As stated in Section 2.1 of the Materials and Methods, we define and differentiate spontaneous fermentations from non-spontaneous fermentations. In the latter case, the non-spontaneous fermentations begin when wort is inoculated with mixed cultures derived from commercial sources, microbes found in non-pasteurized commercially available beer, and/or “house” cultures propagated and maintained by the brewers themselves. In the cases of commercial inocula being used, we provided vendor information in Table 1, and the constituents of these cultures can be found on the vendors’ websites. Unfortunately, for fermentations inoculated with bottle dregs or house cultures, neither we nor the brewers know the identities of the viable microbes present. We apologize for this lack of information, but with samples submitted by citizen-scientists, we were at the mercy of our donors.

Regardless, comparing the samples with known species at the inoculation stage to our final sequencing results is an excellent idea. In the revised manuscript, we now include a Supplementary Table S1 (see attached file) listing the known constituents of the commercial cultures and whether or not they appeared in our deep sequencing results.

  1. (Beer conditions) Actually, microbial communities affected by external factors such as pH, and alcohol contents. Therefore, it need to revise.

Again, this is an excellent point, but as stated above, we can only provide as much information on the cultures, worts, and final beers as was given to us by the sample submitters. Indeed, after sample submission, we surveyed each donor with a variety of questions to gather as much additional brewing and fermentation information as possible. These data were included in the Supplementary Information File S1 as individual tabs in the spreadsheet (noted in the main text at the end of section 2.1). Each tab contains the type of sample, type of brewer, wort recipe, mash profile, original gravity, and exposure to adjuncts, among various other things. In general, for beers produced by spontaneous fermentation, wort was exposed to ambient air, and typical wort conditions were found at fermentation day 1, e.g., pH ranging 4.0 – 5.4 and no alcohol content.

Reviewer 2 Report

I wish you would mention that brewers who use spontaneous fermentation typically don't know what's in their mixed culture instead of just fermentation. It's being nit picky I know, but the examples you use (lambics and American coolships) are spontaneously fermented. 

Some of the figures are a little difficult to read due to the small font. I understand you have limited spacing. However, when printed the font is very difficult to read such as in Figure 7. I wish you pulled out some of the smaller branches and linked it to the area outside of the figure instead of forcing into it. 

I'm not sure if you can change the graph for your bacterial/fungi composition figures. When printed in black and white it's hard to tell the difference between a number of the species.

Author Response

I wish you would mention that brewers who use spontaneous fermentation typically don't know what's in their mixed culture instead of just fermentation. It's being nit picky I know, but the examples you use (lambics and American coolships) are spontaneously fermented.

This is an excellent point, and honestly, the reason we chose to perform this research in the first place. Brewers should know what microbes are in their mixed cultures as well as their fermentations. We changed the text in the Introduction to clarify the point and also highlight this again in the Discussion.  Please see the Track Changes in the edited manuscript for the relevant details.

Some of the figures are a little difficult to read due to the small font. I understand you have limited spacing. However, when printed the font is very difficult to read such as in Figure 7. I wish you pulled out some of the smaller branches and linked it to the area outside of the figure instead of forcing into it.

We again agree with Reviewer 2; some of the figures sized for publication are difficult to read without zooming in. In the updated manuscript, Figures 4B and 7B now include branches connecting organism names in a larger font size. We struggled to make the equivalent changes to Figures 4A and 7A, so instead, we include high-resolution versions of these figures that can be downloaded and magnified as needed as supplementary files. The existence of these files is noted in the figure legends to draw interested readers to them.

I'm not sure if you can change the graph for your bacterial/fungi composition figures. When printed in black and white it's hard to tell the difference between a number of the species.

We tried the experiment, and yes, Figure 2 is difficult to interpret in gray scale. To circumvent this issue, we remade the Figure as shown in the attached file.

Reviewer 3 Report

Dear all,

below are my comments and suggestions

Manuscript ID: fermentation-1329127

The research entitled " Mixed culture metagenomics of the microbes making sour beer " is interesting and useful research in the direction of providing information on the identification of the microorganisms in the sour beers brewed by spontaneous and non-spontaneous methods. However, some changes need to be made in the manuscript itself

Introduction: The introduction can be improved and include more relevant references

Materials and Methods. It was difficult because the text is not understandable in some places. Please verify the method description.

Please indicate the bibliographic sources for Section 2.2.

I recommend moving Table 1 to the Materials and Methods section

The number of bibliographic sources is adequate, more than 80% of the total bibliographic sources are from the last 5 years.

There are some grammatical errors and instances of badly worded/constructed sentences throughout the manuscript. Please refine the language carefully. 

Author Response

Introduction: The introduction can be improved and include more relevant references

We wish that Reviewer 3 had indicated which sections or lines in the Introduction could be improved or how the Introduction could be improved overall, aside from the addition of more references. Lacking that clarification though, we did our best to add new references and information throughout the Introduction. Please see the Track Changes in the attached file for the edits.

Materials and Methods. It was difficult because the text is not understandable in some places. Please verify the method description.

Again, we are happy to comply, but in which places is the text not clear and which method description requires verification? Dr. Bochman, a native English speaker and veteran science copyeditor, completely edited the Methods and Materials section (and the rest of the manuscript) for grammar, spelling, word choice, sentence structure, clarity, etc. Please see the Track Changes for the edits. Hopefully, this has addressed Reviewer #3’s concerns.

Please indicate the bibliographic sources for Section 2.2.

The work described in Section 2.2 was performed by Zymo Research. We now cite their website in this section. Additional bibliographic sources for next-generation sequencing can be found in Section 3 (Results).

I recommend moving Table 1 to the Materials and Methods section

We naively assumed that the Fermentation copyeditors would format the final document to professionally integrate the tables and figures in appropriate positions for publication, but until then, we moved Table 1 to the Materials and Methods section.

The number of bibliographic sources is adequate, more than 80% of the total bibliographic sources are from the last 5 years.

Thank you. We attempted to cite the foundational literature in the field, as well as recent advances.

There are some grammatical errors and instances of badly worded/constructed sentences throughout the manuscript. Please refine the language carefully. 

As state above, Dr. Bochman has completely edited the entire manuscript to comply with the rules of English language grammar and mechanics. Please see the Track Changes for the edits.
